# Learning to Grasp the Ungraspable with Emergent Extrinsic Dexterity

Wenxuan Zhou[1] and David Held[1]

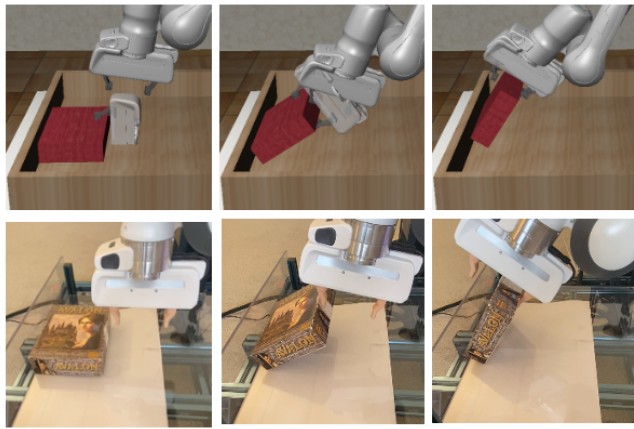

Fig. 1: We study the task of "Occluded Grasping" with extrinsic dexterity. The goal of this task is to reach an occluded grasp configuration (indicated by a transparent gripper attached to the object in the top row). The figure shows the emergent behavior of the trained policy which uses the wall of the bin to rotate the object to reach a grasp.

*Abstract*—A robot can solve more complex manipulation tasks beyond the limitations of its body if it can utilize the external environment such as pushing the object against the table or a vertical wall. These behaviors are known as "Extrinsic Dexterity." Previous work in extrinsic dexterity usually relies on hand-crafted primitives or careful assumptions about contacts. In this work, we explore the use of reinforcement learning (RL) on the extrinsic dexterity with the task of "Occluded Grasping". The goal of the task is to grasp the object in configurations that are initially occluded; the robot must interact with the object and the extrinsic environment to move the object into a configuration from which these grasps can be achieved. To accomplish this task, we train a policy to co-optimize pre-grasp and grasping motions; this results in emergent behavior of pushing the object against the wall in order to rotate and then grasp it. We demonstrate the generality of the learned policy across environment variations in simulation and evaluate it on a real robot with zero-shot sim2real transfer. Videos can be found at **https://sites.google.com/view/grasp-ungraspable.**

## I. INTRODUCTION

Humans have dexterous multi-fingered hands; however, similarly dexterous robot hands are expensive and fragile. Instead, robots can achieve dexterous manipulation with a simple hand by leveraging the environment, known as "Extrinsic Dexterity" [1]. For example, a simple gripper can rotate an object in-hand by pushing it against the table [2], or lifting an object by sliding it along a vertical surface [3]. With the exploitation of external resources such as contact surfaces or gravity, even simple grippers can perform skillful maneuvers that are typically studied with a multi-fingered dexterous hand. Different from a common practice of considering the robot and an object of interest in isolation, extrinsic dexterity focuses on a holistic view of considering the interactions among the robot, the object, and the external environment.

Previous work in extrinsic dexterity has demonstrated a variety of tasks such as in-hand reorientation with a simple gripper, prehensile pushing or shared grasping [1], [2], [3]. However, the underlying approaches come with several limitations such as relying on hand-designed primitives, making assumptions about contact locations and contact modes, or requiring specific gripper design. Instead, we use reinforcement learning (RL) to remove these limitations. With reinforcement learning, the agent can learn a closed-loop policy of how the robot should interact with the object and the environment to solve the task. In addition, when trained with domain randomization, the policy can learn to be robust to different variations of physics. These properties of RL can enable extrinsic dexterity in a more general setting.

We study "Occluded Grasping" as an example of a task that requires extrinsic dexterity. Occluded Grasping is defined with the goal of grasping an object in poses that are initially occluded. Consider, for example, a robot that needs to grasp a cereal box lying on its side on a table; the desired grasp is not reachable because it is partially occluded by the table (Figure 1). To achieve this grasp with a parallel gripper, the robot might rotate the object by pushing it against a vertical wall to expose the desired grasp. This task is in contrast with existing grasping tasks which mostly focus on reaching an unoccluded grasp in free space with a static or near-static scene [4], [5], [6]. Prior work has attempted to design pre-grasp motions of exposing occluded grasp poses with primitives or special gripper design [7]. In our work, the pre-grasp motion is an emergent behavior through a novel reward function that co-optimizes exposing the grasp pose and achieving the grasp pose. In addition, we frame the task as a goal-conditioned RL problem, in which the policy is conditioned on the selected grasp. During training, the policy learns to reach as many grasp poses as possible with an automatic curriculum [8]. During testing, given a set of grasps, the policy can select one of them as a goal to execute.

In summary, we present a system for "Occluded Grasping" as an example of the combination of reinforcement learning and extrinsic dexterity. We provide a comprehensive evaluation of the system both in simulation and on a real Franka Emika Panda robot. We showcase the importance of each components and the generalization of the learned policy across environment variations in simulation and real.

[1]Robotics Institute, Carnegie Mellon University

## II. RELATED WORK

### A. Extrinsic dexterity

"Extrinsic dexterity" is a type of manipulation skills that enhance the intrinsic capability of a hand using external resources including external contacts, gravity, or dynamic motions of the arm [1]. Previous work in extrinsic dexterity has demonstrated complex manipulation tasks with a simple gripper including in-hand reorientation [1], [9], prehensile pushing [2], [10], shared grasping [3], etc. In this work, we study a different task that can further demonstrate the benefit of extrinsic dexterity. Extrinsic dexterity usually involves contact-rich behaviors which poses difficulties in planning and control. Previous work has used hand-crafted trajectories [1], task-specific motion primitives [9], [3] or motion planning over contact mode switches [2], [10], [11], [12]. They come with the restrictions on the contact modes between the finger and the object which will limit the motion and the design of the gripper. In this work, we take an alternative approach of using reinforcement learning to learn a closed-loop policy that considers both planning and control.

### B. Reinforcement Learning for Manipulation

Previous work that uses reinforcement learning for manipulation tasks treat the object and the robot in isolation without considering extrinsic dexterity [13], [14], [8]. In our work, we demonstrate that the agent can benefit from extrinsic dexterity when solving the occluded grasping task.

### C. Grasping

Grasping has been an important task in robot manipulation and has been studied from various aspects.

**Grasp generation:** One area of study in grasping is to generate stable grasp configurations [15], [16], [17], [4], [18], [5], [19]. We assume that we will use the grasps generated by any grasp generation method as input to our system.

**Grasp execution:** To execute a grasp following grasp generation, a motion planner is usually used to generate a collision-free path towards the desired grasp configuration. If there is a set of desired grasps, integrated grasp and motion planning could be considered [20], [21], [6]. [22] uses imitation learning and reinforcement learning to finetune the trajectories from the planner. All of these works aim at achieving the unoccluded grasp configurations in static or near-static scenes. Instead, our work focuses on a complementary direction of achieving occluded grasp locations by interacting with the object of interest.

**Pre-Grasp manipulation:** To deal with occluded grasp configurations, prior work has studied pre-grasps as a preparatory stage [23], [24], [25], [7]. [7] is the most related to our work, but they use a specially designed end-effector to perform the pre-grasp motion and then use a second gripper to grasp the object. We demonstrate that the full grasping task can be solved with a single gripper without special requirements on the end-effector. These previous work typically separates pre-grasp motion and grasp execution into two stages and impose restrictions on the transitions of the stages. In our work, we co-optimize pre-grasp and grasp

execution within an episode without explicit separation of the stages. The pre-grasping behavior emerges through learning without restrictions on object or gripper motions.

**End-to-end grasping:** Another line of work use an end-to-end pipeline for grasping with reinforcement learning [26] or imitation learning [27]. The policy performs an arbitrary grasp of the object without the possibility of specifying a certain set of grasps. Also, there has not been any emergent behavior of exposing occluded grasp pose in existing work.

## III. TASK DEFINITION: OCCLUDED GRASPING

Our work is designed to be used in a pipeline that follows a grasp pose generation method such as [4], [5], [19]. Given a rigid object, we assume a desired grasp $g$ as input to the system. A grasp configuration $g \in SE(3)$ is defined to be the desired 6D pose of the end-effector in the object frame $O$. The grasp is fixed with respect to the object, and it will move when the object moves. On the top row of Figure 1, an example of a desired grasp is shown as a transparent gripper attached to the object. The goal of our work is to learn grasp execution which is to move the end-effector $E$ close to a given $g$ with a pose difference metric $\Delta(g, E)$. In this paper, the task is defined to be successful if the position difference $\Delta T(g, E)$ and the orientation difference $\Delta \theta(g, E)$ are less than the pre-defined thresholds $\varepsilon_T$ and $\varepsilon_P$ respectively at the end of an episode. After successfully reaching the desired grasp pose, the gripper will be closed to complete the grasp. We define an "Occluded Grasping" task to be the case where the grasp $g$ is initially occluded (not in free space). When a set of grasps $G = \{g_i\}$ are available, we may select a grasp $g_i$ from the set $G$ to execute (Appendix VII).

## IV. LEARNING OCCLUDED GRASPING WITH REINFORCEMENT LEARNING

We study the use of reinforcement learning (RL) to train a closed-loop policy for the occluded grasping task defined above. In this section, we will first discuss important design choices of the system considering a single target grasp including the extrinsic environment and the design of the RL problem. Then, we will also discuss how to improve the generalization of the policy using Automatic Domain Randomization [8]. Training and evaluation procedures that process a set of grasps can be found in Appendix VII.

### A. Extrinsic Environment

To showcase the benefits of extrinsic dexterity from object-scene interaction in this task, we construct the scene of the task as having an object in a bin, instead of leaving the object on the table (Figure 2). In Section V, we will show that the emergent policy will utilize the wall of the bin to rotate the object. Without the wall, it is not able to find a strategy that can successfully perform the task.

### B. RL Problem Design

We discuss the design of the RL problem in this section. More details can be found in Appendix I. We train a goal-conditioned policy $\pi(a_t | s_t, g)$ for this task where the goal is a

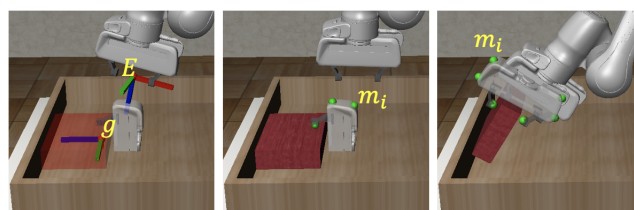

Fig. 2: $E$ denotes the 6D pose of the end-effector. $g$ denotes the target grasp defined in the object frame. Marker locations $m_i$ in green on the target grasp are used to calculate the occlusion penalty.

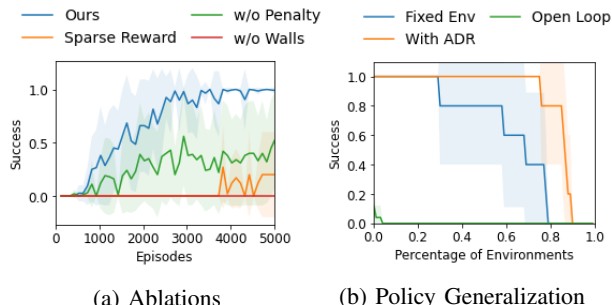

(a) Ablations        (b) Policy Generalization

Fig. 3: **Left:** Ablations on the reward function and the walls. **Right:** Evaluation on the generalization of the policies by sampling 100 environments.

target grasp configuration $g$. $s_t$ includes the pose of the end-effector and the object pose. The action space of the policy is the delta pose of the end-effector $\Delta E$ which will be sent to a low-level Operational Space Controller (OSC). The choice of OSC allows compliant movement for such a contact-rich task (See Appendix I for more discussion). The reward function is designed to co-optimize the pre-grasp motion as well as grasp execution:

$$r = \alpha D(g, E) + \beta \sum_i P(m_i) \tag{1}$$

where

$$D(g, E) = \alpha_1 \Delta T(g, E) + \alpha_2 \Delta \theta(g, E) \tag{2}$$

$\alpha_1$, $\alpha_2$ and $\beta$ are the weights for the reward terms. The first term of Equation 1, $D(g, E)$, is the pose difference between the target grasp and the current end-effector pose. This term is expanded in Equation 2 to include the translational and rotational distance, as described in Section III. The second term of Equation 1 is the target grasp occlusion penalty which is to penalize the gripper if it is occluded by the table. We set several marker points on the *target gripper* (Figure 2) denoted as $m_i$ and compare the height of the markers with the table top. If a marker is below the table top, the height difference will be used as the penalty. Having the occlusion penalty can effectively reduce the local optima where the gripper will reach close to the target grasp (which is occluded) without trying to move the object.

To summarize, the first term of Equation 1 is to optimize for successful grasp execution and the second term is to encourage pre-grasp motions to move the object such that the grasp $g$ becomes unoccluded. An important difference from previous work is that pre-grasp and grasp execution components are optimized together instead of being separated into two stages. We did not have any reward terms that are explicitly related to extrinsic dexterity. In our system, the use of extrinsic dexterity is an emergent behavior of policy optimization given our objective and environmental setup.

### C. Policy Generalization

One benefit of using RL is that it generates a closed-loop policy instead of an open-loop trajectory. A closed-loop policy can ideally generalize to a wider range of state distributions which implies better performance over the

variations of the environment properties such as object size, density, and friction coefficient, etc. The generalization can be improved further by training with domain randomization on the environment variations. This can also benefit sim-to-real transfer. We use Automatic Domain Randomization (ADR) [8] to improve the generalization of the policy. More implementation details can be found in Appendix II.

## V. EXPERIMENTS

### A. Training Curves and Ablations

Details of the experiment setup can be found in Appendix III. In this section, we train the policies with a single desired grasp in the default environment without randomization of the physical parameters. From the training curve shown in Figure 3a, the policy trained with the complete system can reach a success rate of 1 before 4000 episodes which corresponds to 160000 environment steps. We performed an ablations analysis on the design choices to determine which components were the most important to the success of the system. First, we experiment with removing the wall of the bin to evaluate the importance of using the wall for extrinsic dexterity. As shown in Figure 3a, the resulting policy has 0% success rate and pushes the object outside of the table. Second, we performed an ablation on the reward function. When we remove the grasp pose occlusion penalty (the second term of Equation 1), the policy is more likely to get stuck at a local optima of only trying to match the position and orientation of the gripper and thus the average success rate across random seeds becomes lower. An alternative is to use a $\{-1, 0\}$ sparse reward according to the success criteria defined in Section III instead of the reward that we define in Equation 1. With a sparse reward, the policy learns much slower. Training this task with sparse reward makes the exploration task of the policy much more difficult. In addition, ablations on the choice of controller can be found in Appendix V. We also include results for multi-grasp training and multi-grasp selection in Appendix VII.

### B. Emergent Behaviors

Figure 1 shows a typical strategy of the successful policies. The strategy involves multiple stages of contact switches. The gripper first moves close to the object and makes contact

on the side of the object with the left finger. It then pushes the object against the wall to rotate it. During this stage, the gripper maintains a fixed or rolling contact with the object. The object is usually under sliding contact with the wall and the ground of the bin at some of the corners. After the gripper has rotated a bit further and the right fingertip is below the object, the left finger will slide on the object or simply leave the object to let the object drop on the right finger. After the object lies on the right finger, the gripper will try to match the desired pose more precisely. At this point, the policy has executed the grasp successfully and it is ready to close the gripper. We include more visualizations of emergent behaviors in Appendix IV, including another type of successful strategy, local optima behavior and multi-grasp behaviors. Videos can be found on the website [1].

### C. Policy Generalization

In this section, we analyze the performance of the policy across environment variations. The robustness over environment variations might come from the policy being closed-loop and the randomization of the physical parameters during training. Thus, we evaluate over open loop trajectories (*Open Loop*), policies trained over a fixed environment (*Fixed Env*) and policies trained with ADR (*With ADR*). The open loop trajectories are obtained by rolling out the *Fixed Env* policies in the default environment. We also turn off the randomization of the initial gripper pose for *Open Loop*; otherwise, the success rate is too low to compare with even in the default environment. We sample 100 environments from the training range of the ADR policies (Appendix II) and plot the percentage of environments that are above a certain performance metric (Figure 3b). The closed-loop policies are much better than open-loop trajectories across environment variations. The policy trained over a fixed environment is able to generalize to a wide range of variations. With ADR, the generalization can be improved even further. We also modify the important physical parameters one at a time to understand the sensitivity of these parameters in Appendix VI.

### D. Real-robot experiment

To further evaluate the generalization of the policies and demonstrate the feasibility of the proposed system, we execute the policies on the real robot with zero-shot sim2real transfer over 6 test cases shown in Figure 4. There are four box-shape objects with different sizes, density and surface friction. Box-1 has the same size and density as the default object trained in simulation. Box-2 is larger than the training range in the y-direction. Box-3 is larger than the training range in the z-direction. The surface friction are very different for different boxes. For example, Box-3 has tape on its surface which has much higher friction than the others (which can be shown in the videos on the website [1]). However, we do not have access to the true friction coefficients of the objects to compare with the values in simulation. In addition, we evaluate Box-1 with additional

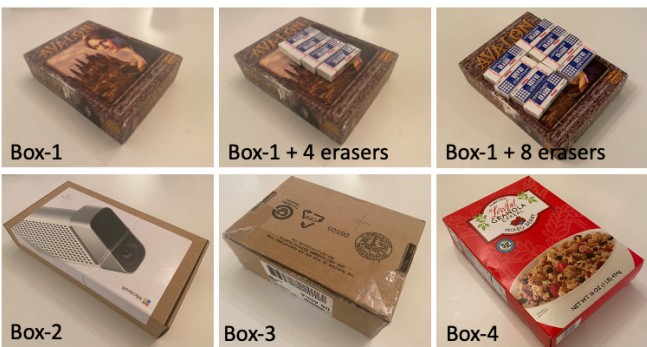

Fig. 4: Test cases for real robot experiments.

TABLE I: Real robot evaluations.

| Object-ID | Size (cm) | Weight (g) | Success w/ ADR | Success w/o ADR |
|---|---|---|---|---|
| Box-1 | (15.0, 20.0, 5.0) | 128 | 10/10 | 10/10 |
| Box-1 + 4 erasers | (15.0, 20.0, 5.0) | 237 | 8/10 | 7/10 |
| Box-1 + 8 erasers | (15.0, 20.0, 5.0) | 345 | 6/10 | 4/10 |
| Box-2 | (15.4, 29.2, 5.8) | 130 | 8/10 | 8/10 |
| Box-3 | (15.3, 22.2, 7.4) | 113 | 10/10 | 4/10 |
| Box-4 | (15.3, 22.2, 7.4) | 50 | 7/10 | 0/10 |
| Average | | | 0.82 | 0.55 |

weights by putting four or eight erasers inside of the box. Note that the erasers will move in the box during execution, which is not modeled in simulation. We evaluate two types of single grasp policies trained in simulation: one policy is trained with Automatic Domain Randomization as described in Section IV-C; another policy is trained on a fixed default environment without domain randomization.

We evaluate 10 episodes for each test case and summarize the results in Table I. Videos of the real robot experiments can be found on the website[1]. Overall, the policy with ADR achieves a success rate of 82% while the policy without ADR achieves 55%. ADR effectively improves the performance over a wider range of object variations. Note that both policies are evaluated on out-of-distribution objects: Box-1 with 8 erasers, Box-3 and Box-4 are out of the training distribution of ADR (See Appendix II); All of the test cases except the first one (Box-1) are out-of-distribution for the policy without ADR. This demonstrates the robustness of the closed-loop policies of the proposed pipeline on such a dynamic manipulation task.

## VI. CONCLUSION

We study the "Occluded Grasping" task of reaching a desired grasp configuration that is initially occluded. With a parallel gripper, the robot has to use extrinsic dexterity to solve this task. We present a system that learns a closed-loop policy for this task with reinforcement learning. In the experiments, we demonstrate that the wall, the choice of controller, and the design of the reward function are all essential components. The policy can generalize across a wide range of environment variations and can be executed on the real robot. One potential extension of our work is to train the policy with a wide variety of object shapes which may require image-based policies. Also, the pipeline can potentially be applied to other extrinsic dexterity tasks.

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

# Appendix I
## More Details of RL Problem Design

**Observations:** We train a goal-conditioned policy $\pi(a_t|s_t, \eta)$ for this task where the goal $\eta$ is a target grasp configuration $g$. Note that the policy only takes one grasp as input but we will discuss how to deal with a set of grasps in Appendix VII. $s_t$ includes the pose of the end-effector in the world frame $^WE$ and the object pose in the world frame $^WO$. We also include the pose of the end-effector in the object frame $^OE = (^WO)^{-1}(^WE)$ because we found that it sometimes speeds up learning. Each pose is represented as a 3D translation vector and a 4D quarternion representation of the rotation. In summary, the input to the policy includes $(g, ^WE, ^WO, ^OE)$ which has a dimension of 28 in total.

**Actions:** An outline of the policy execution pipeline is shown in Figure 5. The action space of the policy is the delta pose of the end-effector $\Delta E$ in its local frame represented by a vector of translation $p \in \mathbb{R}^3$ and a 3D vector of rotation $q \in SO(3)$ with axis-angle representation. Thus, the dimension of the action space is 6. $\Delta E$ and the current gripper pose $E$ form a desired pose $E_d$ at timestep $t$ which will be sent to a low-level Operational Space Controller which will be discussed in the next section.

If the corresponding joint configuration of the desired pose is going to reach joint limits, we will overwrite the policy

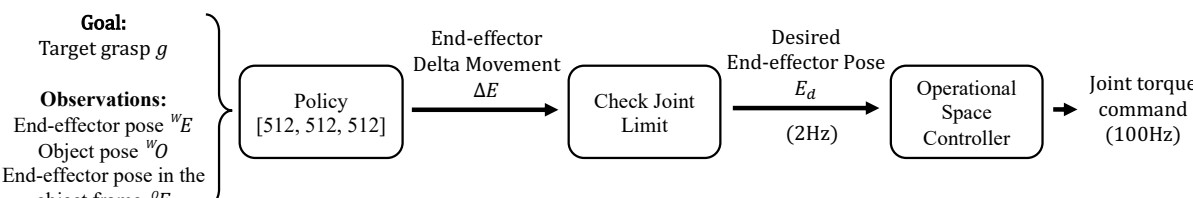

Fig. 5: Outline of policy execution: Given the goal and the observation, the policy outputs a delta movement of the end-effector. If the desired pose is within the joint limit of the robot, it will be sent to the lower level controller.

action to output the desired pose of the previous timestep to the low-level controller. In detail, we use the Jacobian $J$ to estimate the joint configuration of the desired pose:

$$\theta_{joints}^{t+1} = \theta_{joints}^t + J^{-1} \cdot \Delta E \quad (3)$$

where $\theta_{joints}$ are the joint angles. If any joint in $\theta_{joints}^{t+1}$ is close to the limit, the low-level controller will use the previous desired pose instead.

**Low-level controller:** We use Operation Space Control (OSC) as the lower-level controller to achieve the desired pose [28]. Given a desired pose of the end-effector, OSC first calculates the corresponding force and torque at the end-effector to minimize the pose error according to a PD controller with gain $K_p$ and $K_d$. Then, the desired force and torque of the end-effector will be converted into desired joint torques according to the model of the robot. OSC will operate at a higher frequency (100Hz) than the policy $\pi$ (2Hz).

This choice of controller is very important for this task due to the fact that we expect the agent to use extrinsic dexterity to solve the task which involves contacts among the gripper, the object and the bin. There are two benefits of OSC in contact-rich manipulation. First, being compliant in end-effector space allows safe execution of the motions without smashing the gripper on the objects or the bin. Limiting the delta pose and selecting proper gains $K_p$, $K_d$ will limit the final force and torque output of the end-effector. If we use a controller that is compliant in the joint configuration space instead, we will not have direct control over the maximum force the end-effector might have on the object and the bin. Second, as shown in [29], using OSC as the low-level controller might speed up RL training and improve sim2real transfer for contact-rich manipulation.

## APPENDIX II
### DETAILS OF AUTOMATIC DOMAIN RANDOMIZATION

As discussed in Section IV-C, we use Automatic Domain Randomization [8] to improve policy generalization across environment variations. In ADR, the policy is first trained with an environment with very little randomization, and then we gradually expand the variations based on the evaluation performance. For a set of environment parameters $\lambda_i$, each $\lambda_i$ is sampled from a uniform distribution $\lambda_i \sim U(\phi_i^L, \phi_i^H)$ at the beginning of each episode. During training, the policy will be evaluated at these boundary values $\lambda_i = \phi_i^L$ or $\lambda_i = \phi_i^H$. If the performance is higher than a threshold, the boundary value will be expanded by an increment $\Delta$. For example,

if the performance at $\lambda_i = \phi_i^H$ is higher than the threshold, the training distribution becomes $\lambda_i \sim U(\phi_i^L, \phi_i^H + \Delta)$ in the next iteration. Compared to directly training the policy with the entire variations, Automatic Domain Randomization can reduce the need of manually tuning a suitable range of variations for each environment parameter.

Table II summarized the simulation parameters in the experiment. They start from a single initial value and gradually expand to a wider range according to the pre-specific increment step $+\Delta$ on the upper bound and the decrement step $-\Delta$ at the lower bound. We include the final range from ADR expansion in the last column. These ranges are used when we sample 100 environments for evaluation in Section V-C. All the parameters are uniformly sampled from these ranges at the beginning of each episode.

| | Initial Value | $+\Delta$ | $-\Delta$ | Final Range |
|---|---|---|---|---|
| Object size x (m) | 0.15 | 0.01 | -0.01 | [0.14, 0.16] |
| Object size z (m) | 0.05 | 0.01 | -0.01 | [0.04, 0.06] |
| Table friction | 0.3 | 0.1 | -0.1 | [0.1, 0.5] |
| Gripper friction | 3 | / | -1 | [2, 3] |
| Object Density ($g/m^3$) | 86 | 86 | 43 | [43, 172] |
| Action translation scale (m) | 0.03 | / | -0.005 | [0.02, 0.03] |
| Action rotation scale (rad) | 0.2 | / | -0.05 | [0.1, 0.2] |
| Initial distance to wall (m) | 0 | 0.01 | / | [0, 0.02] |
| Table offset x (m) | 0.5 | 0.01 | -0.01 | [0.48, 0.52] |
| Table offset z (m) | 0.07 | 0.01 | 0.01 | [0.055, 0.075] |

TABLE II: Simulation parameters in Automatic Domain Randomization

## APPENDIX III
### EXPERIMENT SETUP

**Simulation:** We build the simulation environment with Robosuite [30] in the MuJoCo simulator [31]. We use a box-shaped object in this task with a default grasp location shown in Figure 1. The object is placed in a bin in front of the robot. We use single grasp training by default; the results related to multi-grasp can be found in Appendix VII. Each episode has a length of 40 timesteps which corresponds to 20 seconds of real time execution. The initial joint configuration of the robot is randomized with a Gaussian of 0.02 rad.

**Real robot experiment:** The policy is trained in the simulator and zero-shot transferred on a physical Franka Emika Panda robot. The code for controlling the robot is built on top of FrankaPy [32]. For real robot experiments, we use Iterative Closest Point (ICP) for pose estimation of the object which matches a template point cloud of the object to the current point cloud [33]. An example of ICP result is shown in Figure 7.

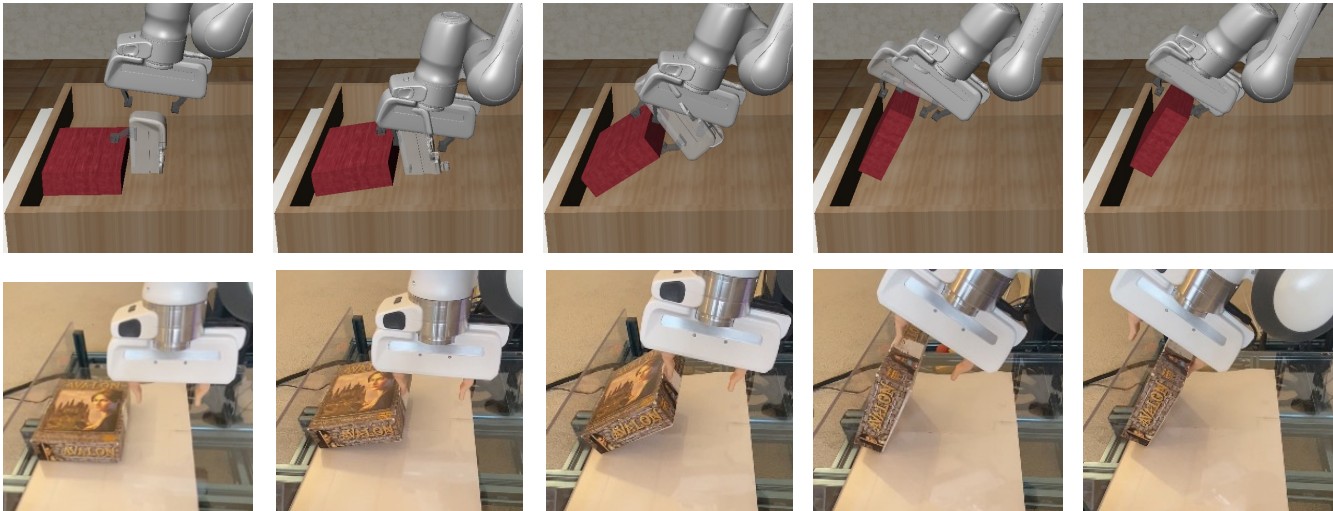

Fig. 6: Emergent behavior of the policy for the occluded grasping task involves multiple stages of contact mode transitions among the gripper, the object and the bin. The figure shows the corresponding stages in simulation versus the real robot execution of the policy.

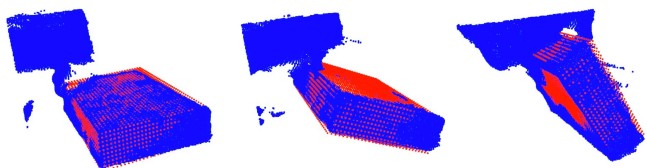

Fig. 7: Illustration of object pose estimation with ICP at three different timesteps of an episode. The blue points are observed point cloud which includes both the gripper and the object. The red points are the template model of the object.

**Evaluation metrics:** We compare the policies across 5 random seeds of each method and plot the average performance with standard deviation across seeds. Our main evaluation metric is the success rate at the final step of the episode computed as $\mathbb{1}(\Delta T < 3\,cm) \cdot \mathbb{1}(\Delta\theta < 10\,deg)$ (See Section III for definitions). We use 10 episodes for each evaluation setting.

**Implementation details:** We use Soft Actor Critic [34] to train the RL policy with the impementation from rlkit. Both the policy network and the Q-function are parameterized as a multi-layer perceptron (MLP) with 3 layers of 512 neurons.

## APPENDIX IV
### ADDITIONAL RESULTS ON EMERGENT BEHAVIORS

In Section V-B, we discuss a typical emergent strategy of solving this task as a result of the design of the full system. Figure 6 includes a more detailed view of this strategy across multiple stages in simulation and on the real robot.

One of the key decisions in this strategy is to use the left finger to rotate the object instead of the right finger. One might suppose an alternative approach which is to use the right finger to scoop the object against the wall and then directly roll the finger underneath the object to reach the

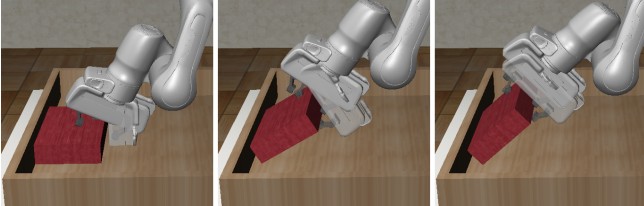

(a) **Local optima:** The gripper uses the right finger to lift the object and get stuck at a local optima.

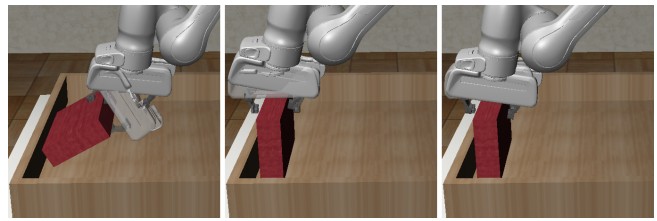

(b) **Standing object:** One of the successful strategies is to flip the object until it stands on the side and then reach the grasp.

Fig. 8: More visualizations on the emergent behavior of the policies.

grasp. However, this strategy is not physically feasible on the parallel gripper due to the limited degree of freedom of the finger. We observe that the policies that follow this strategy during exploration usually get stuck at a local optima without successfully reaching the grasp (Figure 8a).

Another type of successful strategy from some of the seeds is to flip the object to stand on its side and then move to the grasp (Figure 8b). This strategy overfits to the box object because it relies on the fact that the object remains stable after the flip. If the agent is trained on a more diverse set of objects without such stable poses, it might learn to avoid this strategy; however, for a box object, this is also a viable approach.

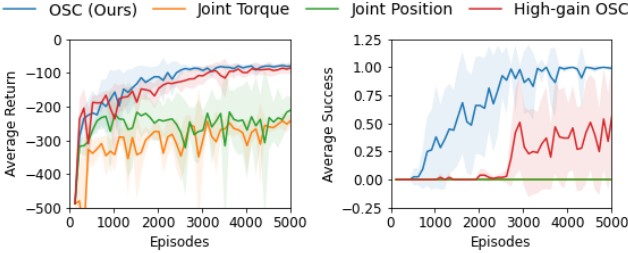

Fig. 9: Ablations on the choice of controller.

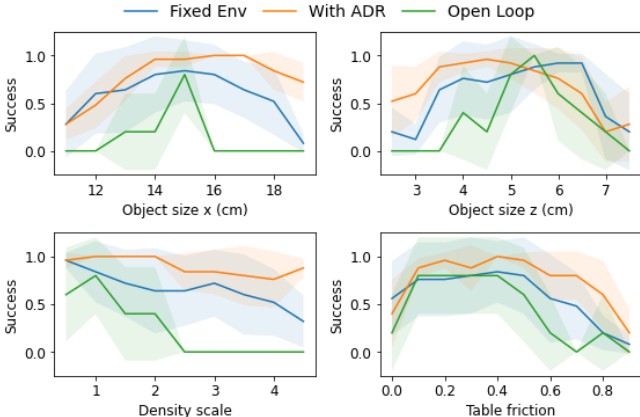

Fig. 10: Evaluation on the generalization of the policies by changing one physical parameter at a time.

## APPENDIX V
## ABLATIONS ON LOW-LEVEL CONTROLLER

We compare our method to different types of controllers to demonstrate that the choice of Operational Space Controller (OSC) is critical for extrinsic dexterity. From Figure 9, both joint torque and joint position control lead to worse performance which indicates the importance of using end-effector coordinates for the action space. We also try increasing the gain of OSC so that it becomes roughly equivalent to position control. The success rate becomes lower which demonstrates that being compliant is important for the success of contact-rich tasks in addition to the importance of compliance for safety considerations.

## APPENDIX VI
## MORE RESULTS ON POLICY GENERALIZATION

To further analyze the robustness of the policy across environment variations, we modify the important physical parameters one at a time to understand the sensitivity of the policies to these parameters. Following Section V-C, we include the comparison of open loop trajectories (*Open Loop*), policies trained over a fixed environment (*Fixed Env*) and policies trained with ADR (*With ADR*). The closed-loop policies with ADR can deal with much wider variations of physical parameters than open loop trajectories.

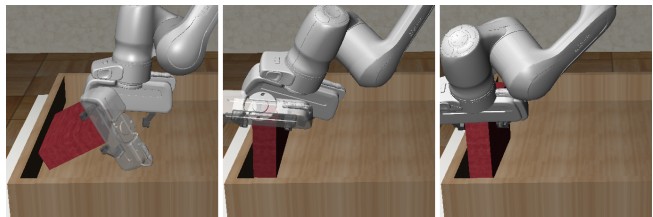

(a) **MultiGrasp-Front:** When the desired grasp is at the corner, the policy flips the object by pushing it on the side and then move close to the grasp.

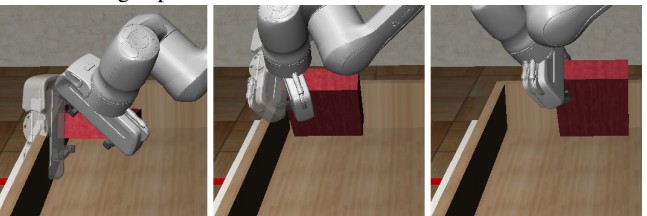

(b) **MultiGrasp-Side:** The policy can use another side of the wall to rotate the object and reach the desired grasp.

Fig. 11: Visualizations of the multi-grasp policies.

## APPENDIX VII
## MULTIGRASP TRAINING AND SELECTION

In previous sections, we only consider the scenario when a single grasp is given for each episode. In this section, we consider the scenario in which a set of desired grasp configurations $G = g_i$ are given. We will first discuss the method for multi-grasp training and selection and then provide the experimental results.

**MultiGrasp Training with Curriculum:** During training, we aim at covering as many grasp configurations from $G_{train}$ as possible. The straight-forward approach is to uniformly sample a goal $g \sim G_{train}$ for each episode. However, previous work has shown that learning directly over such a diverse set of goals might create a difficulty for policy learning [35], [36]. Instead, we use an automatic curriculum following [8] to gradually expand the set of grasps to be trained with. We start the training with just a single fixed grasp; after the policy has achieved a success rate larger than a threshold, it will be trained on a slightly larger set with grasps close to the initial grasp location.

**MultiGrasp Selection:** During testing, a set of grasps $G_{test}$ is provided. Our method selects the best grasp within the set that maximizes the learned Q-function for the current observation: $g^* = \arg\max_{g \sim G_{test}} Q(s_t, a_t, g_t)$. Selecting the best grasp from the set (instead of just using a single grasp) can improve the performance of the grasping task, following previous work in integrated grasp and motion planning [20], [21], [6]. The learned Q-function can select the grasp that is most easily reached with the trained policy; which grasp is selected thus depends both on the environmental configuration as well as how well the policy has learned to achieve different grasp configurations.

**MultiGrasp Training Results:** In this experiment, we train the policy to reach a range of grasp locations with curriculum as described above. Given the box object, we generate the grasp configurations around the box and pa-

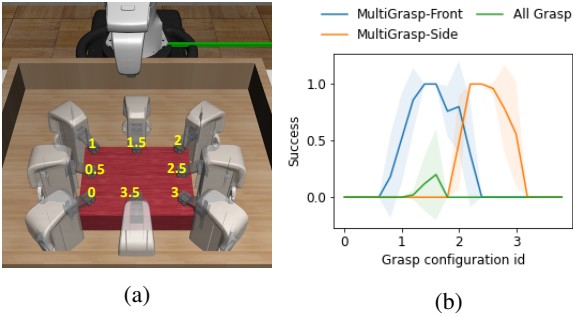

(a)

(b)

Fig. 12: Multi-grasp training: **Left:** Visualization of the range of grasp configurations and the grasp IDs used in multi-grasp training. **Right:** Performance of the multi-grasp policies across grasp configurations.

rameterized the grasps into a continuous scalar grasp ID in the range of $[0,4]$ (Figure 12a). Grasp ID 1.5 is the default grasp we use in the single grasp experiments. The policy is trained with an automatic curriculum. When the success rate of policy on a boundary case of the training range is above 0.8, it will expand the range of grasps by 0.25. For example, if the policy is currently training with grasps $[1,2]$, and the success rate evaluated at grasp ID 1 is above 0.8, the new training range will be $[0.75,2]$. We train two types of multi-grasp policies starting from two different grasp poses: *MultiGrasp-Front* which starts the training from ID 1.5 and *MultiGrasp-Side* which starts the training from ID 2.5. As a baseline, we also train a policy by uniformly sampling from the entire set of grasps without using ADR, named *All Grasp*.

Figure 11a and Figure 11b include qualitative examples of the behaviors of *MultiGrasp-Front* and *MultiGrasp-Side*. The policy will rotate the object first and then try to match the pose more precisely. *MultiGrasp-Side* will use a different wall of the bin to rotate the object than *MultiGrasp-Front*. Figure 12b shows the performance of these policies evaluated over across grasp configuration IDs. We found that both *MultiGrasp-Front* and *MultiGrasp-Side* are able to expand from a single grasp to most of the grasps on one side of the object based on the curriculum. The policies have difficulties in reaching other sides potentially due to exploration issues or limited policy capacity. It may require a completely different strategy to reach different grasp configurations (Figure 11) which is difficult to learn with a single policy (related to [35]). In contrast, *All Grasp* has difficulties in learning any of the grasp configurations, thus showing the importance of using a curriculum for multi-grasp training.

**MultiGrasp Selection Results:** To compare grasp selection methods, at the beginning of each episode, we sample 50 grasp configurations from the training range of the policy. The grasp selection methods will use it as the set of desired grasps. We evaluate the following grasp selection options:

- *ArgmaxQ*: passes all the possible grasp configurations into the Q-function and select the one that corresponds to the highest Q-value.
- *PoseDiff*: selects the grasp by the closest distance to the current gripper location according to Equation 2 (with

TABLE III: Comparison of grasp selection methods in two scenarios: front grasps and side grasps. When grasping from the side, the policy achieves better performance when using the Q-function to select the grasp.

| | **MultiGrasp-Front** | **MultiGrasp-Side** |
|---|---|---|
| ArgmaxQ | $1.00 \pm 0.00$ | $1.00 \pm 0.00$ |
| ArgmaxQ-$t_0$ | $1.00 \pm 0.00$ | $1.00 \pm 0.00$ |
| PoseDiff | $1.00 \pm 0.00$ | $0.96 \pm 0.08$ |
| PoseDiff-$t_0$ | $1.00 \pm 0.00$ | $0.50 \pm 0.43$ |
| Uniform | $0.54 \pm 0.16$ | $0.90 \pm 0.06$ |

the same weights as the reward function).

- *ArgmaxQ-$t_0$*: selects the grasp according to *ArgmaxQ* only during the first timestep of the episode instead of selecting it at every timestep.
- *PoseDiff-$t_0$*: selects the grasp according to *PoseDiff* only during the first timestep of the episode instead of selecting it at every timestep.
- *Uniform*: samples a grasp from the set uniformly.

The result is summarized in Table III. For *MultiGrasp-Front*, all of the methods other than *Uniform* achieve 100% success rate. In this case, the best grasp according to the Q-function does correspond to the grasp that is the closest to the gripper at grasp ID 1.5. For *MultiGrasp-Side*, *ArgmaxQ-$t_0$* has higher success rate than *PoseDiff-$t_0$*. The policy has a more complicated maneuver to reach the side grasp so the Q-function may capture the difficulty of the goal better than pose difference. At the beginning of the episode, the Q-function selects ID = 2.5 while the pose difference selects ID = 2. If we keep this goal throughout the episode, *PoseDiff-$t_0$* has a much lower success rate than the other baselines. If the policy can select the goal throughout the episode instead (*PoseDiff*), the performance can be improved compared to *PoseDiff-$t_0$*.