# OpenReview forum: "Learning to Grasp the Ungraspable with Emergent Extrinsic Dexterity"
_ICRA.org/2022/Workshop/Contact-Rich — ICRA 2022 Workshop: RL for Manipulation Poster_

### Official Review · Reviewer_RDoZ · 2022-05-05

**Rating:** 7
**Confidence:** 4

**Review:**

### Summary
This paper presents a reinforcement learning-based approach to solve "occluded grasping" with extrinsic dexterity. The main idea is to learn to grasp an object, with the desired grasp that is initially occluded. To reach such desired grasp, the robot needs to exploit the environment to manipulate the object such that the desired grasp becomes available. The authors proposed to use RL to learn such behavior.

The topic is very interesting and relevant to the workshop.

### Comments
- This paper provides a well-organized and clear introduction to the proposed method.

### Suggestions
- Although it is acceptable and in many cases necessary to simplify a task to demonstrate the benefits of a particular and very specific approach, it would be better if the task was not too simple. In this paper, the task is to grasp a box in a desired way (target grasp pose) that is occluded in some way. The box is always in front of the robot and placed next to the wall. The advantage of such a simple environment is that a blind state (just object pose and robot state) is enough to stumble into the behavior that can solve the task, such as moving close to the object and then rotating it by pushing against the wall. The disadvantage is that the RL agent would not generalize to very similar environments, such as having the wall diagonal to the box. How long would it need to be trained (re-trained) to stumble until finding a satisfactory behavior?
- Tackling a more complicated environment would make this work stand out more. For that, the RL agent would benefit also from better observations of the environment. Having an estimation of the position of the wall, even if a rough noisy one, would be a great opportunity to compare your proposed method with planning-based methods. Would your method provide a more robust solution to uncertainty than traditional/previous methods?
- Consider discussing the limitations and if possible potential approaches to address such limitations.

---

### Official Review · Reviewer_qURK · 2022-05-08
**Review of Paper "Learning to Grasp the Ungraspable with Emergent Extrinsic Dexterity"**

**Rating:** 7
**Confidence:** 4

**Review:**

This paper studies the use of model-free Reinforcement Learning to solve tasks requiring extrinsic dexterity. As an example, the authors considered the "Occluded Grasping" task, in which the desired grasping pose is initially unreachable. The authors showed that emergent extrinsic dexterity could be learned through the appropriate design of reward function and action space. A sim-to-real experiment was conducted, validating the proposed method on the physical robot.

**Strengths**
- The emergent Extrinsic Dexterity behavior is interesting and natural through RL problem design (the existence of the wall and reward function design).
- The experiments are comprehensive. The paper has put effort to compare different design choices such as the action space, reward function, whether to do domain randomization, etc.
- The sim-to-real experiment results are good.

**Weaknesses**
- I doubt that the second reward term in equation (1) is generalizable to different tasks that require extrinsic dexterity as this reward term is designed specifically for the considered problem (the flat table). One way to address this is to exploit the collision checking function that is available in most physics simulators. Here, the reward can be set to the overlapping distance between the gripper and an external object if a collision occurs between them and 0 ortherwise.